# Magnetic Multiwall Carbon Nanotube Decorated with Novel Functionalities: Synthesis and Application as Adsorbents for Lead Removal from Aqueous Medium

**Ghadir Hanbali** [1], **Shehdeh Jodeh** [1,*], **Othman Hamed** [1,*], **Roland Bol** [2], **Bayan Khalaf** [1], **Asma Qdemat** [3], **Subhi Samhan** [4] **and Omar Dagdag** [5]

1   Department of Chemistry, Faculty of Science, An-Najah National University, P.O. Box 7, Nablus 00001, Palestine; g.hanbali@najah.edu (G.H.); bayan.kh107@hotmail.com (B.K.)
2   Institute of Bio and Geosciences, Agrosphere (IBG-3), Forschungszentrum Jülich GmbH, 52425 Jülich, Germany; r.bol@fz-juelich.de
3   Jülich Center for Neutron Science, Peter Grunberg Institute, Forschungszentrum Jülich GmbH, 52425 Jülich, Germany; qdemat@fz-juelich.de
4   Palestine Water Authority, Ramallah 00001, Palestine; subhisamhan@yahoo.com
5   Laboratory of Agroresources, Polymers and Process Engineering (LAPPE), Department of Chemistry, Faculty of Science, Ibn Tofail University, BP 133, 14000 Kenitra, Morocco; omar.dagdag@uit.ca.ma
*   Correspondence: sjodeh@hotmail.com (S.J.); ohamed@najah.edu (O.H.); Tel.: +970-599-590-498 (S.J.); +970-594-466-271 (O.H.)

**Abstract:** Water pollution is one of the major challenges facing modern society because of industrial development and urban growth. This study is directed towards assessing the use of multiwall carbon nanotube, after derivatization and magnetization, as a new and renewable absorbent, to remove toxic metal ions from waste streams. The adsorbents were prepared by, first oxidation of multiwall carbon nanotube, then derivatizing the oxidized product with hydroxyl amine, hydrazine and amino acid. The adsorbents were characterized by various techniques. The adsorption efficiency of the multiwall carbon nanotube adsorbents toward Pb(II) was investigated. The effect of adsorbent's dose, temperature, pH, and time on the adsorption efficiency were studied and the adsorption parameters that gave the highest efficiency were determined. The derivatives have unique coordination sites that included amine, hydroxyl, and carboxyl groups, which are excellent chelating agents for metal ions. The thermodynamic and kinetic results analysis results indicated spontaneous adsorption of Pb(II) by the multiwall carbon nanotube adsorbents at room temperature. The adsorption process followed pseudo-second-order and Langmuir isotherm model. The adsorbents were regenerated using 0.1 N HCl.

**Keywords:** magnetic multiwall carbon nanotube; adsorption; kinetics; isotherm; thermodynamic; lead

## 1. Introduction

Metal water pollution is a major environmental concern as some are highly toxic elements, even at low concentrations [1]. The toxicity of heavy metals is an inevitable consequence as they accumulate in the soft tissues of the human body [2,3]. These metals might enter the human body through different ways, among these is drinking water, and unlike organic pollutants, heavy metal ions are nonbiodegradable.

Heavy metal ions exists in waste streams of various industries, including leather tanning, metal plating, radiator manufacturing, batteries manufacturing, mining operations, smelting, alloy industries, and fertilizer industries [4].

Lead is considered one of the most toxic metals, it has access to the human body via ingestion, inhalation or skin assimilation [5]. Once inside the body, lead is absorbed and stored in bones, blood, and tissues [6]. Lead can cause severe damage to the brain, kidneys and, ultimately, it could lead to death. By simulating calcium, lead can cross the blood-brain barrier and destroys myelin sheaths of neurons, decreases their numbers, inhibits with neurotransmission routes and reduces the growth of neurons [7].

In nature, lead can compete with other minerals in and on plants surfaces and may inhibit photosynthesis, and negatively effect on plant growth and survival at sufficiently high concentrations [8].

The bioaccumulation of lead in food sources and its toxicity to biological systems attributable to increased concentrations over time has called for a significant pressure for removal. Several technologies have successfully removed lead from polluted water; among these are adsorption, membrane filtration, ion exchange bioremediation, precipitation and coagulation, and heterogeneous photocatalyst [9–11]. Among the most effective technologies, adsorption has been proven the highest removal efficiency at a reasonable cost. In addition, adsorbents can be regenerated and reused [12,13]. An unlimited number of effective adsorbents were developed and reported in the literature. Among the most effective metal adsorbents is multiwall carbon nanotube (MWCNT), it has shown significant success and has been employed in many of commercial applications [14]. Several published studies reported that carbon nanotubes are among the most effective in water treatment and adsorption of heavy metals such as copper, lead, cadmium. It also showed high efficiency in polar and nonpolar organic molecules. This was related to the various physical and chemical interactions MWCNT can afford in the from $\pi$–$\pi$ interactions, covalent bonding, hydrogen bonding [15], and electrostatic interactions [16].

Chemical modification or doping is believed to optimize MWCNT properties for a certain specific application. A common method is to enhance the dispersion, and optimize the use of multi-wall carbon nanotubes through chemical functionality. This enables the formation of chemical interconnection between MWCNT and targeted materials. Both non-covalent and covalent structures have been employed to improve solubility and physical properties of MWCNT [17].

MWCNT functionalization with P, O, and N containing groups on the surface of MWCNT allow the researchers to control the surface area, surface charge, hydrophobicity/hydrophilicity, dispersion. and improve adsorption capacity and selectivity towards the heavy metals [18,19]. For instance, Li et al. [20] showed that acid-refluxed carbon nanotubes (CNTs) have enhanced ability to adsorb Pb(II) ions from water. Wang et al. [21] showed that adding oxygen functional groups on acidic MWCNT improved the adsorption capacity of lead ion. Tofighy and Mohamadi [22] reported that the adsorption on oxidized CNTs sheets followed the order: $Pb^{2+} > Cd^{2+} > Co^{2+} > Zn^{2+} > Cu^{2+}$. Wang et al. [23] showed that oxidized MWCNT caused a rise in the zeta potential value resulting in the formation of negative surface charge because of the creation and ionization of functional groups (–COOH &–OH). It has been observed that the adsorption of Pb(II) onto acidic MWCNT was not uniform and mainly collected on the cap and defective sites of the MWCNT as adsorbent.

During recent years, the focus has been on magnetic adsorbents to avoid problems associated with adsorption regeneration. The main advantages of magnetic composites include: high strength, strong adsorption rate and improved adsorption ability [24]. The composite adsorbent could effectively achieve the separation of solid–liquid under an active magnetic field without filtration or centrifugation. This results from the properties of the magnetism, which simplifies post-processing. In addition, magnetic recycling can help prevent nano-adsorbents from occurring in the natural environment, and mitigate future hazards [25].

This work focused on a MWCNT grafted with various functionalities as a potential new adsorbent for toxic metal ions. The MWCNT was first oxidized then converted to acid chloride. MWCNT functionalized with acid chloride was reacted separately with hydroxylamine, cystine, and hydrazine.

The prepared grafted MWCNT were magnetized and then evaluated as an adsorbent for Pb(II) removal from water. The effect of various factors on the adsorption efficiencies of grafted MWCNT were evaluated.

The novelty of this work can be summarized as the grafted MWCNTs presented in this work represent the first example in the oxidized MWCNT literature modified with such functionalities and applied for removal of Pb(II). In addition, previous studies on lead removal using a single functional group provided lower removal than the one presented in this study.

## 2. Materials and Methods

Chemicals and MaterialsAll chemicals used in this work were purchased from Sigma-Aldrich (Jerusalem, Israel) and used as received.

### 2.1. Instrument and Characterization

Infrared spectra were recorded using (FTIR-SHIMADZU, Shimadzu Corporation, Kyoto, Japan, Model: FTIR-8700). The following parameters were used: resolution 4 $cm^{-1}$, spectral range 600–4000 $cm^{-1}$, number of scans 128. Flame atomic absorption spectroscopy (iCE 3300, Thermo Fisher Scientific, Cambridge, UK) was used to determine lead ion concentration. Raman spectra were recorded on Bruker RFS 100 FT-Raman spectrometer, scanning electron microscopy ((Hitachi SX-650 machine (Tokyo, Japan)) was used to study the surface morphology or the adsorbents, Brunauer−Emmett−Teller surface area analysis (Micromeritics, Norcross, GA, USA), Vibrating Sample Magnetometer (VSM-LAKESHORE 7404, Boston, MA, USA). All results of the characterization can be found in our previous study [26].

### 2.2. Magnetization of Modified Multiwall Carbon Nanotube

The procedure includes four steps, at the beginning, MWCNT was oxidized then the developed carboxylic acid functionality was converted to acid chloride by reacting it with oxalyl chloride. Then the acid chloride was converted to hydroxamate, hydrazine and amino acids by reacting it with hydroxyl amine, hydrazine and cystine, respectively (1).

In the first step, MWCNT (0.1 g) was treated with 100 mL of 69% $HNO_3$ in a flask of 500 mL. The flask was vibrated in an ultrasonic bath for 30 min at 25 °C. Next, the mixture was diluted with deionized water to reach 400 mL and then filtered through a polycarbonate membrane (0.22 μm). The same procedure was repeated exactly with $H_2O_2$ (30% v/v) instead of $HNO_3$. Hydrogen peroxide was used to complete the oxidation process, but mildly. The pH of the filtrate was roughly 7.0 by washing the solid with deionized water, then the product was dried under a 24 h vacuum to produce the carboxylic acid-functionalized MWCNT (MWCNT-COOH) [27].

Oxidized MWCNT (MWCNT-COOH) (0.1 g), were stirred in 2 mL of oxalyle chloride in the presence of 2–3 drops of dimethylformamide (DMF), and 2 mL of triethyl amine (TEA) at 70 °C for 24 h under $N_2$. After cooling to room temperature, the excess TEA was washed repeatedly with anhydrous tetrahydrofuran (THF) and then dried at 70 °C under vacuum in order to remove any traces of adsorbed TEA on the surface of acylated MWCNT. This samples are labelled as MWCNT-COCl [28].

Three separates of solutions of hydroxylamine (0.2 g), cystine (0.5 g) and of hydrazine (200 uL) in 1 mL THF and 0.5 mL pyridine was prepared. To each solution, w 0.05 g of MWCNT-COCl was added. The mixtures were stirred for 30 min at room temperature, then refluxed at 100 °C for 96 h. The residual hydroxylamine, cystine, and hydrazine were removed by rinsing with ethanol and sonication for 15 min. This rinsing process was repeated three times, until a clear ethanol was produced. The remaining solid was suspended in dichloromethane, sonicated and centrifuged. The collected black solid was dried under vacuum and labelled as magnetized multiwall carbon nanotube functionalized by hydroxyl amine (MWCNT-HA), magnetized multiwall carbon nanotube functionalized by cysteine (MWCNT-CYS), and magnetized multiwall carbon nanotube functionalized by hydrazine (MWCNT-HYD) [29].

A mixed solution of 0.1 M ferric chloride hexahydrate ($FeCl_3 \cdot 6H_2O$) and 0.05 M ferrous chloride tetrahydrate ($FeCl_2 \cdot 4H_2O$) was prepared with one to two molar ratios and then mixed with functionalized MWCNT and suspended for 2 h.

To precipitate iron oxides, 5.0 M $NH_4OH$ solution was added dropwise until pH adjusted at 10 at 70 °C, and then kept under continuous stirring for 1 h. The solid was allowed to cool and magnet separate, and then rinsed with distilled water and ethanol. The composite obtained was dried for 2 h in a furnace at 100 °C. This samples are labelled as (m-MWCNT-HA, m-MWCNT-CYS, m-MWCNT-HYD) [30].

### 2.3. Adsorption Study

All adsorption runs were carried out in plastic vials (50-mL) that were placed in a water bath equipped with a thermostat and a shaker. The effect of various variables such as metal ion concertation, pH values, adsorbent dosage, adsorption time, and temperature on adsorption efficiency were studied. The adsorption experiments were performed on lead (II) ion. After every adsorption run, a clear supernatant sample from the adsorption mixture was collected via magnet and analyzed by a flame atomic adsorption analysis at 193.7 nm to determine the residual lead (II) concentration. All adsorption experiments were carried out in triplicate, and the mean was reported. The adsorbent efficiency and the amount of lead (II) ions adsorbed that was adsorbed ($q_e$, mg/g) was calculated by the Equation (1):

$$q_e = \frac{V(Co - Ce)}{W} \tag{1}$$

where $V$ is the volume of the solution (L), $C_o$ is the initial lead concentration (mg·L$^{-1}$), $C_e$ is lead concentration at equilibrium (mg·L$^{-1}$), and $W$ is the adsorbent's mass (g).

### 2.3.1. Adsorption Isotherms Models

Adsorption isotherms models are important for the description of the interaction mechanism between adsorbate and adsorbent. Several models are available for this purpose, among these, the most common are Langmuir and Freundlich. These models take into account essential comparative guidelines with small distinction in their approaches [31].

The Langmuir model adopts the formation of a monolayer of adsorbate on a homogeneous surface of an adsorbent and expressed as:

$$\frac{C_e}{q_e} = \frac{1}{q_m}C_e + \frac{1}{q_m K_L} \tag{2}$$

where $C_e$ is the equilibrium concentration of adsorbate (mmol·L$^{-1}$), $q_e$ is the amount of adsorbate adsorbed per unit weight of adsorbent (mmol·g$^{-1}$), $q_m$ is the adsorption capacity (mmol·g$^{-1}$), or monolayer capacity, and $K_L$ is a constant (L·mmol$^{-1}$).

Langmuir isotherm can be identified by constant dimensions separation factor ($R_L$) as shown by the following equation [32]

$$R_L = \frac{1}{(1 + K_L C_o)} \tag{3}$$

where $C_o$ is the highest initial concentration of adsorbate (mg·L$^{-1}$), $K_L$ (L·mg$^{-1}$) is Langmuir constant.

The value of the $R_L$ refers to the form of isotherm to be either unfavorable ($R_L > 1$), linear ($R_L = 1$), favorable ($0 < R_L < 1$) or irreversible if ($R_L = 0$).

The Freundlich isotherm describes the adsorption between the adsorbates and the adsorbents with a heterogeneous surface. The rate of adsorption/absorption varies according to the degree of energy at the adsorptive sites. Freundlich's equation is expressed as shown in Equation (3):

$$\ln Q_e = i\frac{1}{n}\ln iC_e + lnK_F \tag{4}$$

where $K_F$ (mmol/g) and 1/n are the constant characteristics of the system [32].

$K_F$ is an indicator of adsorption capacity of the adsorbent and 1/n is an indicator of favorability of adsorption process. If (10 > n > 0) this donates a favorable adsorption process.

Langmuir and Freundlich's isotherms could be used to define the relationship between the amounts of Pb(II) adsorbed on m-MWCNT-HA, m-MWCNT-CYS, m-MWCNT-HYD adsorbents and its equilibrium concentration in solutions.

### 2.3.2. Adsorption Kinetics

Numerous adsorption kinetic models have been set to define kinetics and rate-determining steps. These models give evidence about the performance of the adsorption system and the rate at which a specific component is removed using a specific adsorbent. Besides, it determines whether the adsorption process is physical or chemical in nature and which step is the rate-determining step. Examples of the adsorption kinetic models include pseudo-first-order, intra-particle diffusion kinetic model, pseudo-second-order models, intra-particle diffusion kinetic model, first-order reversible reaction model, Elovich's model, etc. [33].

Pseudo-first-order kinetics are developed for describing adsorption kinetics, and are considered as the earliest model. The equation for this model can be written as follows:

$$\log(q_e - q_t) = \log q_e - \left(\frac{K_1}{2.303}\right)t \tag{5}$$

where $q_e$ and $q_t$ are the masses of the adsorbate in equilibrium or at time t per unit mass of adsorbent (mg g$^{-1}$). k$_1$ is the rate constant of the first-order pseudo-adsorption model (mg·g$^{-1}$·min$^{-1}$).

The graph of log $(q_e - q_t)$ as a function of t gives a straight line for first-order pseudo adsorption with log q$_e$ as Y intercept [34].

The pseudo-second-order models are primarily based on the idea that the step to determine may be chemical adsorption, which involves exchanging or sharing electrons between adsorbate and adsorbent.

The net equation for pseudo-second-order can be addressed as an equation:

$$\frac{t}{q_t} = \frac{1}{q_e}t + \frac{1}{K_2 q_e^2}\,i \tag{6}$$

where $K_2$ is the equilibrium rate constant of the adsorption pseudo-second order (g·mg$^{-1}$ min$^{-1}$).

The graph of t/$q_t$ versus t should give a linear relationship that allows the calculation of a second order rate constants, $K_2$ from the Y intersect and $q_e$ from the slope [30,35,36].

The metal ions adsorption is mainly achieved through three steps, including migration of metal ion from liquid to surface of the adsorbents (film diffusion process), diffusion of metal ions within the porous structure (intraparticle diffusion process, and metal ion adsorption on the surface to adsorbent). In general, the third step isn't time-consuming and is not considered as a rate-controlling step.

Intra-particle diffusion kinetic model suggested by Weber and Morris. The net equation of this kinetic model is:

$$q_t = K_{id}t^{0,5} + C \tag{7}$$

where $K_{id}$ is the constant diffusion rate (mg·g$^{-1}$min$^{-1/2}$). C is a constant representing the thickness of the boundary layer (mg g$^{-1}$). A graph of $q_t$ with respect to t$^{0.5}$ will show a linear relationship with the constant C as Y intersect. When the curve moves through the origin, adsorption is controlled by the process of inter-particle diffusion. Otherwise, it is dominated by film diffusion [37].

Activation energy can also be measured by an Arrhenius equation:

$$\text{Ln } K_2 = \text{Ln A} - \frac{\text{Ea}}{\text{RT}} \tag{8}$$

In general, if $E_a$ between 5 kj/mol to 40 kj/mol the adsorption is physisorption whereas value (40 kj·mol$^{-1}$ to 800 kj·mol$^{-1}$) refers to chemisorption.

### 2.3.3. Adsorption Thermodynamics

The thermodynamic study is done by ascertaining enthalpy, free energy, entropy.

Thermodynamic parameters are required to determine whether the process is spontaneous or not. Gibbs free energy change, $\Delta G^o$, is an indication of the spontaneity of a chemical reaction and therefore is an important criterion for spontaneity. Both enthalpy ($\Delta H^o$) and entropy ($\Delta S^o$) factors must be considered to determine Gibb's free energy of the process. Reactions occur automatically at a certain temperature if $\Delta G^o$ is a negative quantity.

The following equation is the general equation that Connect between thermodynamics parameters

$$\Delta G^o = \Delta H^o - T\Delta S^o \tag{9}$$

where T is the absolute temperature (K).

The change in Gibbs energy can be expressed by the following equation:

$$\Delta G^o = - RT \ \ln \ K_D \tag{10}$$

where, $K_D$ is the constant of thermodynamic equilibrium equal to ($q_e/c_e$) with a unit of (L·g$^{-1}$). R is the gas constant, 8.314 J·mol$^{-1}$K$^{-1}$.

The net equation of the last two equations can be expressed as the follows:

$$\ln K_d = -\frac{\Delta H^o}{RT} + \frac{\Delta S^o}{R} \tag{11}$$

The plot of In$K_d$ against (1/T) gives a line with ($\Delta S^o$ /R) as the Y-intersect and ($-\Delta H^o$/R) as the slope. The resulting graph is identified as a Van't Hoff diagram [38].

### 2.3.4. Regeneration of Adsorbents

The adsorbent was separated by magnet, washed with 0.1 N HCl (10 mL) solution to release adsorbed metal ions, and then washed with distilled water. It was then left to dry at room temperature for 24 h. The three adsorbents were treated in the same manner.

### 2.4. Real Water Sample

The actual sample was used to study adsorbents efficacy for removal heavy metals. A real water sample was collected from Jericho city. The initial and final concentrations of the metals were measured using ICP/MS (Elan 9000 model, PerkinElmer, Inc., Waltham, MA, USA). A 0.02 g of adsorbent was mixed in 50 mL plastic vials with 10 mL of water sample under maximum condition; the amount adsorbed was calculated by comparing the concentrations of the metals before and after the adsorption.

## 3. Results and Discussion

### 3.1. Adsorbents Analysis

MWCNT was oxidized following a published procedure [26]. The developed carboxylic acid functionality was converted to acid chloride by reacting it with oxalyl chloride as shown in Scheme 1. Then the acid chloride was converted to hydroxamate, hydrazine and amino acids by reacting it with hydroxyl amine (HA), hydrazine (HYD) and cystine (CYS), respectively. The presence of these functionalities was confirmed by FTIR, which shows that when MWCNT-COOH was modified by HA, HYD, and CYS, several new peaks appeared in the spectra. Based on the literature: the peak at 1640 cm$^{-1}$ is due to C=O stretching vibration. C–N stretching vibration and N–H bending vibration

appear at 1550 cm$^{-1}$ (Figure S1). Raman spectroscopy shows the spectra of MWCNT, oxidized MWCNT, and devised as MWCNT. It could be clearly noticed that the D-band of MWCNT-COOH shifted from 1360 to 1349 cm$^{-1}$ and G-band from 1605 to 1600 cm$^{-1}$ and this shows that a large defection was created after oxidation. Additionally, the ratio of the intensity of the D-band peak to that of the G-band peak (ID/IG) of the pure MWCNT was 1.33, while that for the MWCNT–functionalized with HA, CYS, HYD was 1.71 (Figure S2). The images of scanning electron microscopy of all samples show disordered and highly porous morphologies. The unmodified MWCNT showed less agglomeration than others (Figure S3). In addition, chemical modification leads to removal of the ends of MWCNT, which increases surface area as confirmed by Brunauer−Emmett−Teller (BET) (Table 1). The magnetization of prepared MWCNT confirmed by VSM, which shows that the magnetization curve for MWCNT sample has a semi-linear representation, which indicates a paramagnetic state while magnetization curve for m-MWCNT-HA, m-MWCNT-CYS, and m-MWCNT-HYD, showed a sigmoidal response with no hysteresis, which is an indication of the presence of a saturated superparamagnetic component in the samples (Figure S4). All characteristic techniques confirmed that HA, CYS and HUD were successfully drafted on MWCNT, and also created functionals, which are expected to have excellent activities toward metal ions [39].

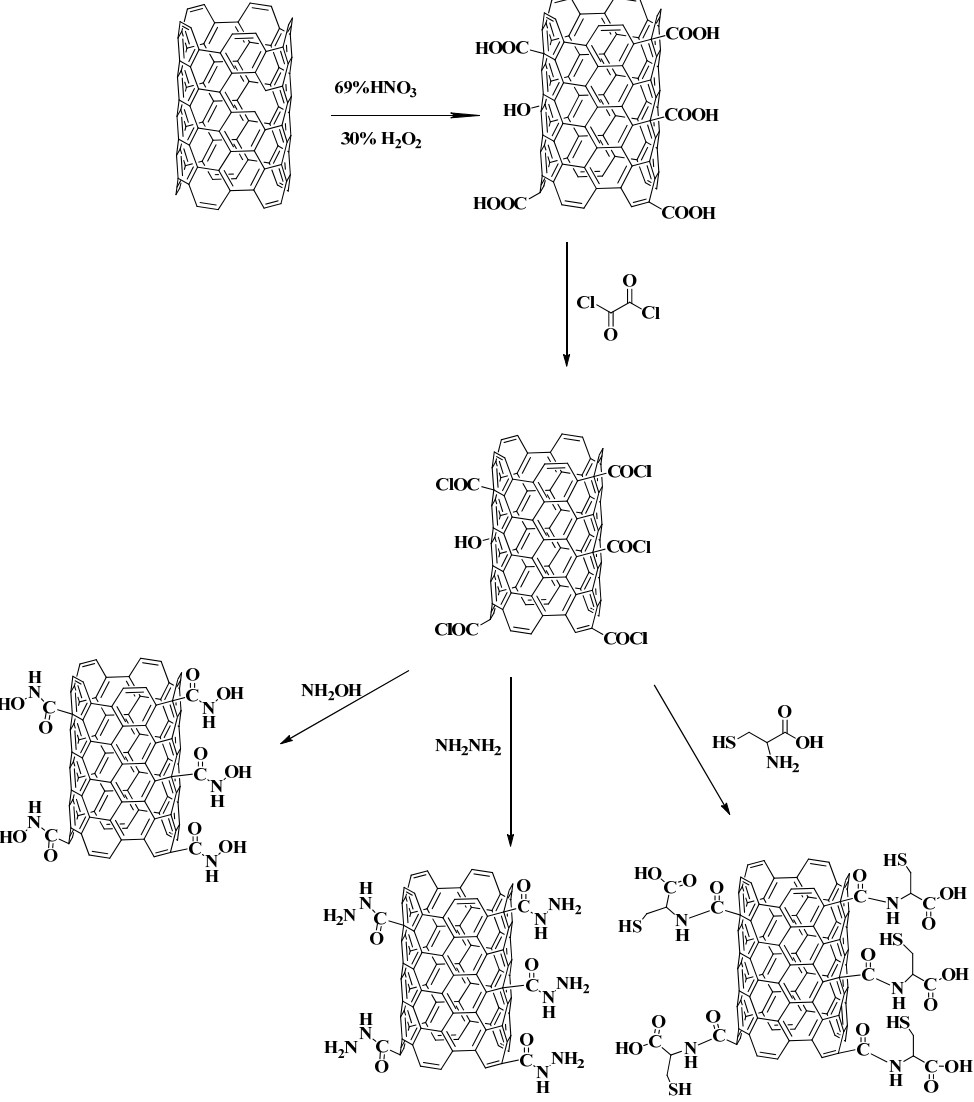

**Scheme 1.** Preparation of magnetic functionalized multiwall carbon nanotube.

With reference to previous studies, there are several studies related to oxidation and surface modification of MWCNT. The novelty of this work can be summarized as the grafted MWCNTs presented in this work comprise the first example in the literature of oxidized MWCNT modified with such functionalities and applied for lead removal. Additionally, previous studies on the removal of lead using a single functional group provided lower removal than that presented in this study (Table 1). Other studies with higher removals than that presented in this study were based on using photodegradation or membranes, which are extremely expensive methods compared with the method presented in our study.

**Table 1.** Previous studies on lead removal from water.

| Adsorbents | Maximum *Adsorption* Capacity ($qm$) or % Removal | Conditions | Surface Area ($m^2$/g) | References |
|---|---|---|---|---|
| MWCNTs | 1 | pH: 5, CNTs dosage: 0.05 g, Temp:280–321 K | 134 | [40] |
| MWCNTs AC | 4 18 | pH: 5, Temp:298–323 K, CNTs dosage:1 g, Contact time: 20–120 min | 162.16 1124.8 | [41] |
| Oxidized CNTs | 104.0 | pH: 7, Temp: 298 K, $C_0$: 100–1200 | 66 | [42] |
| CNTs | 62.5% | pH: 5, CNTs dosage: 0.05 g, Temp: 298 K, $C_0$: 5–60 | 98.6 | [43] |
| Oxidized MWCNTs | 97.08% | pH: 6–11, CNTs dosage: 0.05–0.3 g | 75.4 | [44] |
| m-MWCNT-HA m-MWCNT-CYS m-MWCNT-HYD | 99.8% 97.4% 97.5% | pH: 8, Temp: 298 K, $C_0$: 10 ppm | 151 154.5 187 | Our research |

### *3.2. Lead Ion Adsorption*

The adsorption was carried out using a batch process. During the prior process, a known weight of adsorbent (0.02 g) was suspended in an aqueous solution of lead ion, separated by a magnet, and analyzed. The analysis was performed on the filtrate to determine the amount of the residual lead ions. The effects of several variables such as contact time, adsorbent dosage, temperature, and pH were evaluated to determine the best conditions for the highest adsorption efficiency. The adsorption study was performed on Pb(II) ions using (m-MWCNT-HA), (m-MWCNT-CTS), (m-MWCNT-HYD) adsorbents. Each experiment was repeated three times, and the average was used when we analyzed the data.

### 3.2.1. Effect of Contact Time

This experiment was performed to determine the optimal adsorption time. A 10 mL solution of Pb(II) with 10 ppm concentration was placed in a vial and shaken with 0.02 g of an adsorbent for a varied period of time; ranging from 1 to 120 min.

Adsorption curves are presented in Figure 1a. The figure shows that in the beginning, the development of monolayer was very rapid and the maximum removal was attained after approximately 30 min for all three adsorbents. After that, the adsorption leveled off and became nearly constant.

The high percentage of removal of Pb(II) in the beginning was due to the presence of the vacant sites which decrease steadily with time passing. In general, excellent adsorption efficiency for Pb(II) was exhibited by the three adsorbents.

The same type of interaction appears when using m-MWCNT-HA and m-MWCNT-CYS.

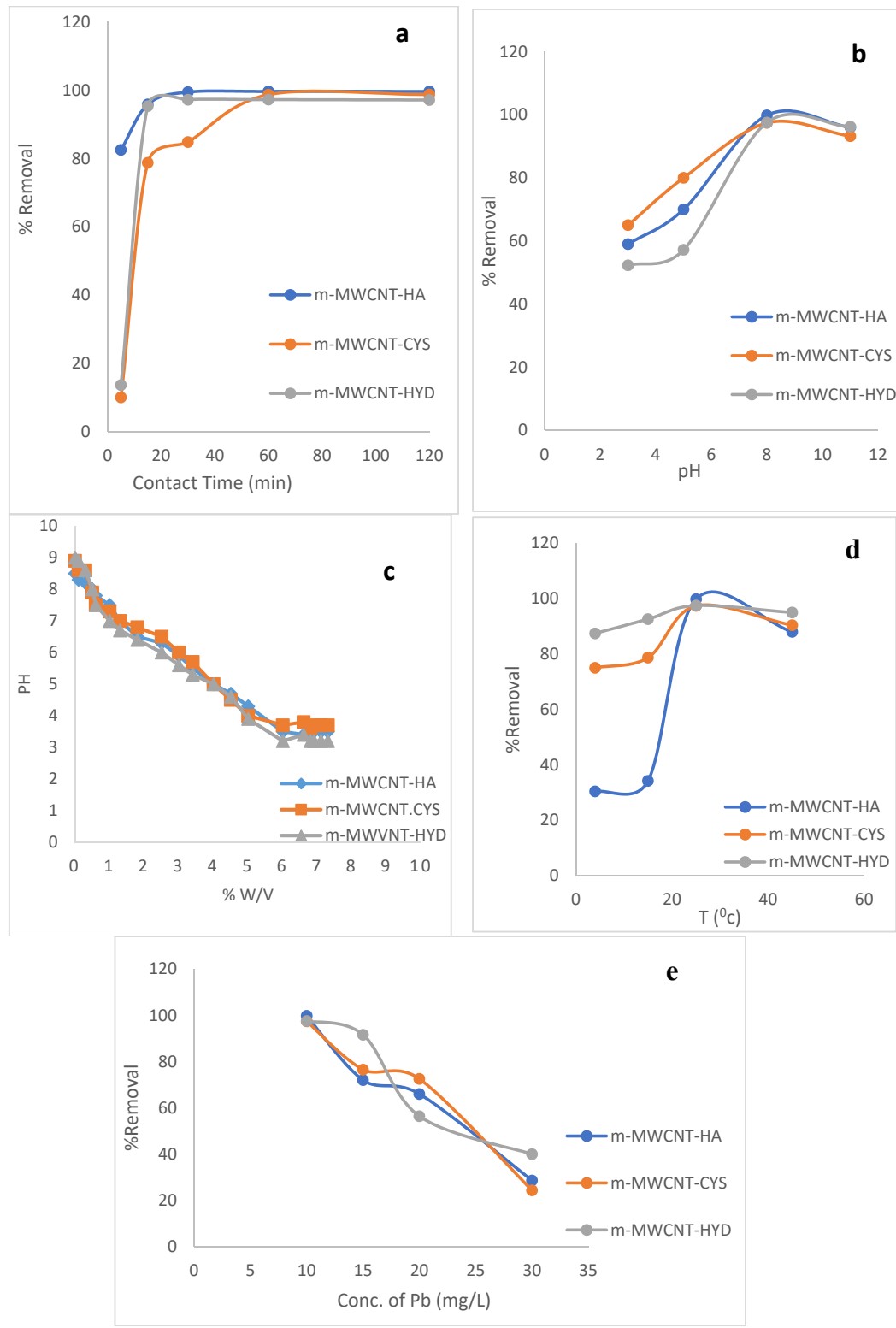

**Figure 1.** Effect of (**a**) contact time, (**b**) pH, (**c**) experimental mass titration curves, (**d**) temperature, (**e**) Pb(II) initial concentration on magnetized multiwall carbon nanotube functionalized by hydroxyl amine (m-MWCNT-HA), magnetized multiwall carbon nanotube functionalized by cysteine (m-MWCNT-CYS), and magnetized multiwall carbon nanotube functionalized by hydrazine (m-MWCNT-HYD).

### 3.2.2. Effect of pH

The effect of pH on the adsorption efficiency is shown in Figure 1b. The pH effect on adsorption is an important factor as it affects the surface charge of the adsorbents. At pH values of 3.0 or less, the amine functionality becomes ammonium whereas, at high pH value over 5.5, the amine is not protonated and the carboxylic acid functionality becomes carboxylate. The highest adsorption efficiency was observed at a pH 8.0 then it started to decline. At pH value of about 8.0, the functional groups present in the adsorbents (amine and carboxyl) carry a lone of pair of electrons causing them to behave as a strong chelating agent.

The pH effect can also be explained in terms of $pH_{pzc}$ of adsorbents. In $pH_{pzc}$ the positive and negative groups are equal and thus sum surface charges become zero.

Mass titration technique (MT) was used to determine zero-point charge (pzc) [45]. When observing Figure 1c, it can be investigated that pzc equals 3.5, 3.7, 3.2 for m-MWCNT-HA, m-MWCNT-CYS, m-MWCNT-HYD, respectively, with the highest adsorption value appeared at $pH > pH_{pzc}$. Consequently, electrostatic attractions among negatively charged surface of adsorbent and metal-ions could occur and influence to sorption sorbent mass (Figure 2).

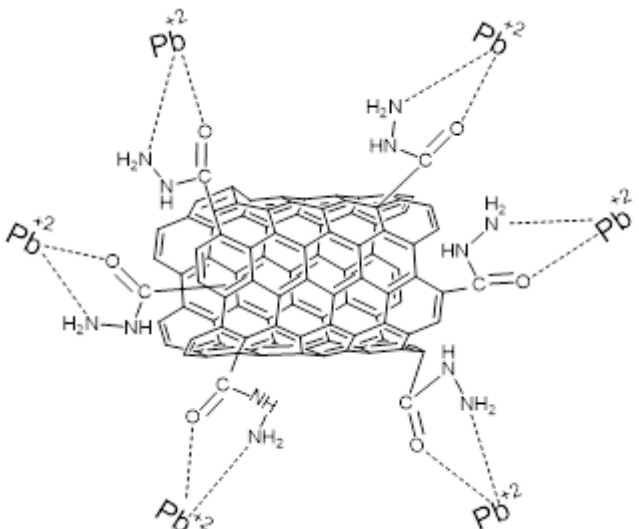

**Figure 2.** Mechanism of lead adsorption on m-MWCNT-HYD.

### 3.2.3. Effect of Temperature on Lead Adsorption

Adsorption equilibrium is reached when the chemical potential of the adsorbed solute and the chemical potential the solute in aqueous solution are equal. This equilibrium is temperature-dependence since the chemical potential is a function of temperature.

In this study, the dependence of adsorption equilibrium of Pb(II) on m-MWCNT-HA, m-MWCNT-CYS, and m-MWCNT-HYD was evaluated using different temperatures (5–50 °C) at a pH value of 8. The effect of temperature on the percentage of lead adsorption is shown in Figure 1d, as shown in the figure the maximum adsorption was at room temperature for the three adsorbents.

### 3.2.4. Effect of Lead Initial Concentration

The results of Pb(II) initial concertation on adsorption efficiency is shown Figure 1e. The highest removal percentage of Pb(II) for the three adsorbents at 10.0 ppm concentration.

The results demonstrate that at lower concentrations, the ratio of the number of metal ions to the existing sorption sites is low and thus the adsorption under these conditions is independent of the initial concentration. However, as the concentration of Pb(II) increases, the sites available for adsorption become less, and therefore, the removal of metals becomes highly dependent on the initial

concentrations. The results show that the rate of Pb(II) ion removal decreases as the initial concentration of metal ions increases.

### 3.3. Equilibrium Isotherm Models for Lead Adsorption

Langmuir and Freundlich's isotherms were applied to describe the relationship between amounts Pb(II) adsorbed on m-MWCNT-HA, m-MWCNT-CYS, and m-MWCNT-HYD adsorbents and its equilibrium concentration in solutions, results are presented in Figure 3a,b.

Freundlich and Langmuir adsorption isotherms parameters were calculated. The values of $R^2$ using Langmuir adsorption isotherm are approximately one. The value of $R_L$ was found to be. 0.002, 0.012, 0.007 for m-MWCNT-HA, m-MWCNT-CYS, m-MWCNT-HYD, respectively, and this indicates that Langmuir adsorption isotherm is favorable.

From Freundlich isotherm data shown in Table 2, the value of $R^2$ is much smaller than 1 (0.15, 0.02, 0.26) for m-MWCNT-HA, m-MWCNT-CYS, m-MWCNT-HYD, respectively), this indicates that the adsorption is not suited with Freundlich isotherm.

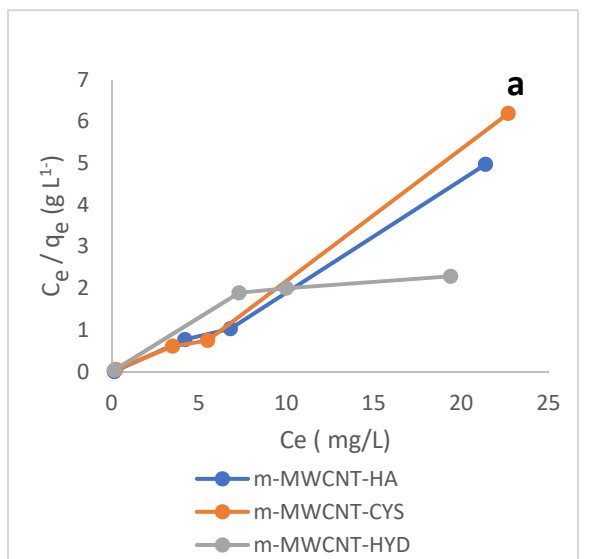 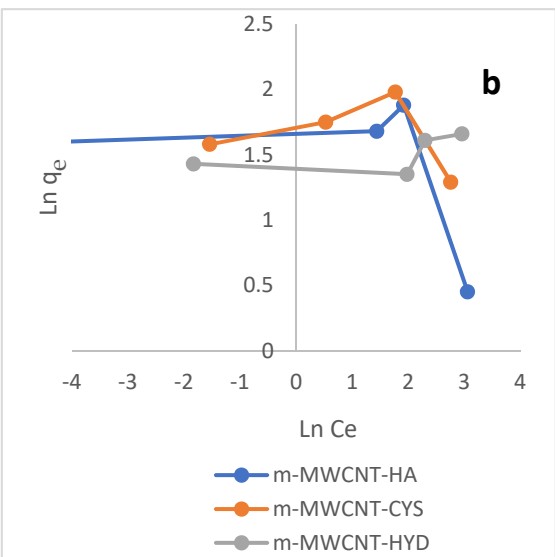

**Figure 3.** (**a**) Langmuir isotherm plot, and (**b**) Freundlich isotherm plot for Pb(II) adsorption.

**Table 2.** The Langmuir and Freundlich parameters for Pb(II) adsorption on m-MWCNT-HA, m-MWCNT-CYS, and m-MWCNT-HYD.

|  | Langmuir Isotherm | | | | Freundlich Isotherm | | |
|---|---|---|---|---|---|---|---|
|  | $q_m$ (mg·g$^{-1}$) | $K_L$ (g·L$^{-1}$) | $R_L$ | $R^2$ | $K_F$ (mg·g$^{-1}$) | n (L·mg$^{-1}$) | $R^2$ |
| m-MWCNT-HA | 1.15 | 39.13 | 0.002 | 0.99 | 4.22 | $-12.80$ | 0.15 |
| m-MWCNT-CYS | 3.31 | 0.76 | 0.012 | 0.98 | 5.31 | $-41.60$ | 0.02 |
| m-MWCNT-HYD | 9.09 | 0.20 | 0.007 | 0.72 | 4.30 | 28.50 | 0.26 |

### 3.4. Adsorption Kinetic Model

To study the mechanism of the adsorption process, pseudo first-order, pseudo second-order and intra-particle diffusion kinetic models were evaluated.

Adsorption kinetic models are represented in Figure 4. The kinetic study revealed that, the adsorption of Pb(II) on m-MWCNT-HA, m-MWCNT-CYS, m-MWCNT-HYD follows the pseudo-second-order kinetic mechanism. The correlation coefficient $R^2$ and $q_e$ experimental and theoretical which in this kinetic model are approximately one (Table 3).

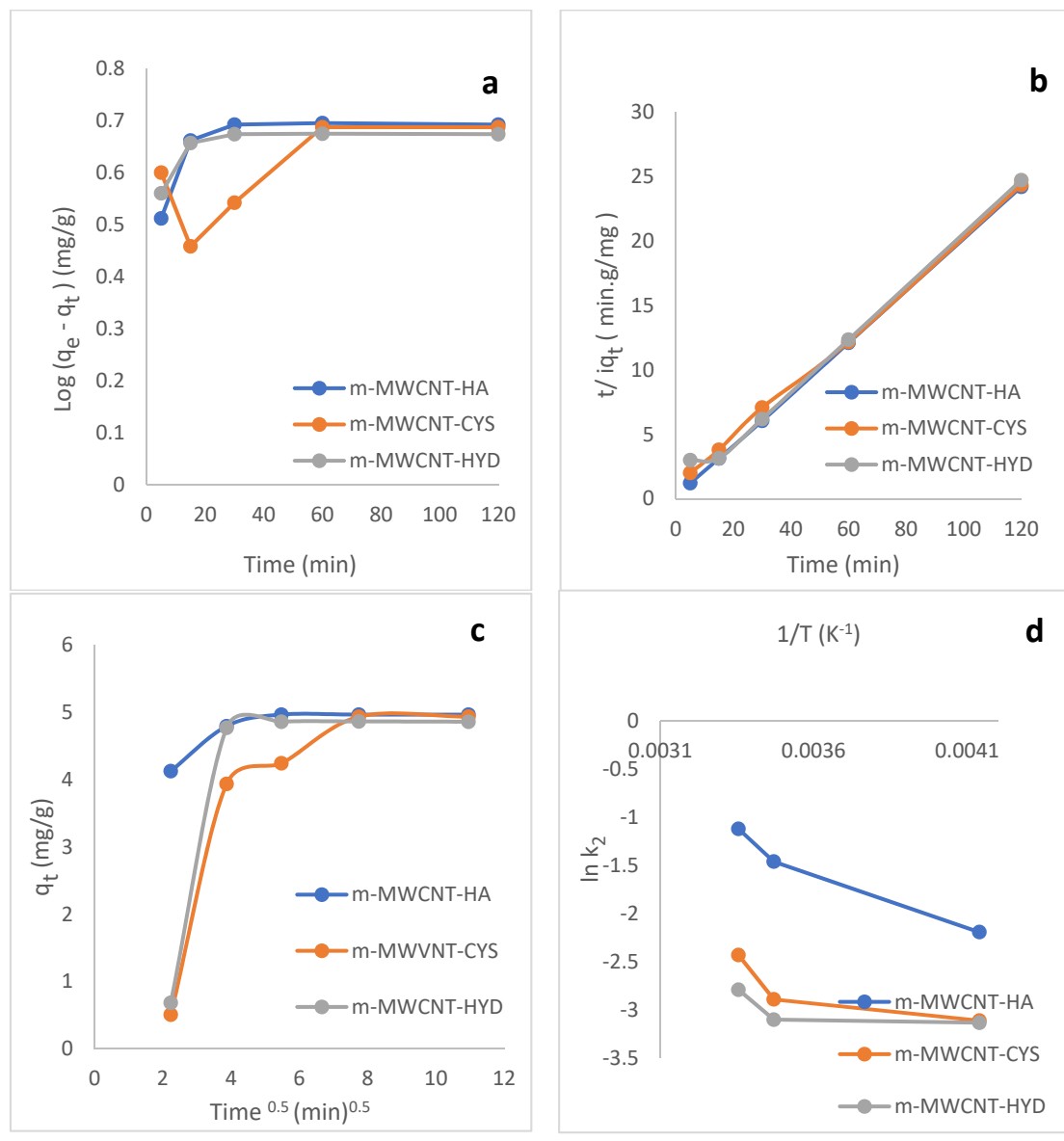

**Figure 4.** (**a**) Pseudo first-order plot. (**b**) Pseudo second-order plot. (**c**) Intra-particle diffusion plot. (**d**) Arrhenius plot for the adsorption of Pb(II) on m-MWCNT-HA, m-MWCNT-CYS, and m-MWCNT-HYD.

**Table 3.** Parameters for Pb(II) adsorption of kinetic models of pseudo-first order, pseudo-second and intraparticle diffusion on m-MWCNT-HA, m-MWCNT-CYS and m-MWCNT-HYD.

| | Pseudo First-Order Kinetics | | | | Pseudo Second-Order | | | |
|---|---|---|---|---|---|---|---|---|
| | $q_e$ | $q_e$ | $K_1$ | | $q_e$ | $K_2$ | $E_a$ | |
| | exp | (mg g$^{-1}$) Calc | (mg g$^{-1}$ min$^{-1}$) | $R^2$ | (mg g$^{-1}$ min$^{-1}$) | (mg g$^{-1}$ min$^{-1}$) | (kj) | $R^2$ |
| m-MWCNT -HA | 4.96 | 4.03 | −0.0023 | 0.33 | 4.99 | 0.32 | 10.50 | 0.99 |
| m-MWCNT-CYS | 4.93 | 3.35 | −0.0034 | 0.51 | 5.15 | 0.03 | 5.65 | 0.99 |
| m-MWCNT-HYD | 4.85 | 4.16 | −0.0013 | 0.32 | 5.39 | 0.04 | 2.50 | 0.91 |
| | **Intra-Particle Diffusion** | | | | | | | |
| | C (mg g$^{-1}$) | | $K_{id}$ (mg g$^{-1}$min$^{-0.5}$) | | | $R^2$ | | |
| | 4.29 | | 0.07 | | | 0.51 | | |
| | 1.172 | | 0.41 | | | 0.60 | | |
| | 1.90 | | 0.34 | | | 0.40 | | |

In the intraparticle diffusion plot shown in Figure 3, the curve is not passing through the origin, so the adsorption is controlled by the external diffusion, and the interparticle diffusion mechanism.

The $E_a$ value was calculated form Arrhenius equation by plotting ln $K_2$ versus 1/T (Figure 4d). The obtained value of Ea indicates that the adsorption processes of Pb(II) ions by the three adsorbents m-MWCNT-HA, m-MWCNT-HYD and m-MWCNT-CYS are physical.

### 3.5. Adsorption Thermodynamic

The Van't Hoff plot was used to calculate the common thermodynamics parameters: $\Delta S^o$, $\Delta H^o$, and $\Delta G^o$ for Pb(II) adsorption on m-MWCNT-HA, m-MWCNT-CYS, m-MWCNT-HYD as shown in Figure 5.

As shown in Table 4, the values of free energy were negative. This sign is a clear indication that the adsorption is spontaneous and favorable while the values of enthalpy were positive. This is a clear indication of the endothermic process. The positive $\Delta S^o$ values reveal an increase in randomness in the solid and liquid surface, indicating the accumulation of Pb(II).

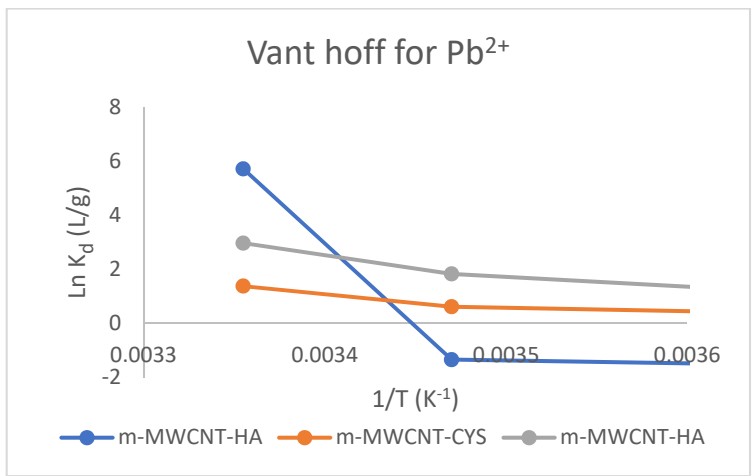

**Figure 5.** Van't Hoff plot for Pb(II) adsorption on m-MWCNT-HA, m-MWCNT-CYS, and m-MWCNT-HYD.

**Table 4.** Thermodynamic parameters for the adsorption of Pb(II) on m-MWCNT-HA, m-MWCNT-CYS, and m-MWCNT-HYD.

|  | $\Delta H^o$ (kj) | $\Delta S^o$ (J K$^{-1}$) | $\Delta G^o$ (25 °C) |
|---|---|---|---|
| m-MWCNT-HA | 205.60 | 0.72 | −10.15 |
| m-MWCNT-CYS | 17.26 | 0.06 | −2.11 |
| m-MWCNT-HYD | 50.60 | 0.19 | −6.90 |

As shown in Table 4, the adsorption of Pb(II) on m-MWCNT-HA, m-MWCNT-CYS, m-MWCNT-HYD endothermic and spontaneous process ($\Delta H^o > 0$) and spontaneous ($\Delta G^o < 0$).

### 3.6. Adsorbent Regeneration

The adsorbents regeneration and reuse as absorbents for Pb(II) was evaluated. The three adsorbents were tested in three cycles, m-MWCNT-HA maintained its efficiency toward Pb(II), while the other two adsorbent m-MWCNT-CYS and m-MWCNT-HYD showed some reduction in the efficiency in the third cycles (Figure 6). To remediate the MWCNTs after regeneration, future studies will be needed.

Biodegradation of CNTs by enzymes was studied by Allen et al. (2008). They reported that oxidized single-wall CNTs (SWCNTs) could be degraded by enzymatic oxidation with horseradish peroxidase (HRP), a plant enzyme. They also reported that degradation took place within 10 days

using hydrogen peroxides. This degradation will be leading to production of oxidized polyaromatic hydrocarbons and ultimately $CO_2$ [46]. Serra et al., studied the circular zero-residue process using microalgae for efficient water decontamination, biofuel production, and carbon dioxide fixation [47].

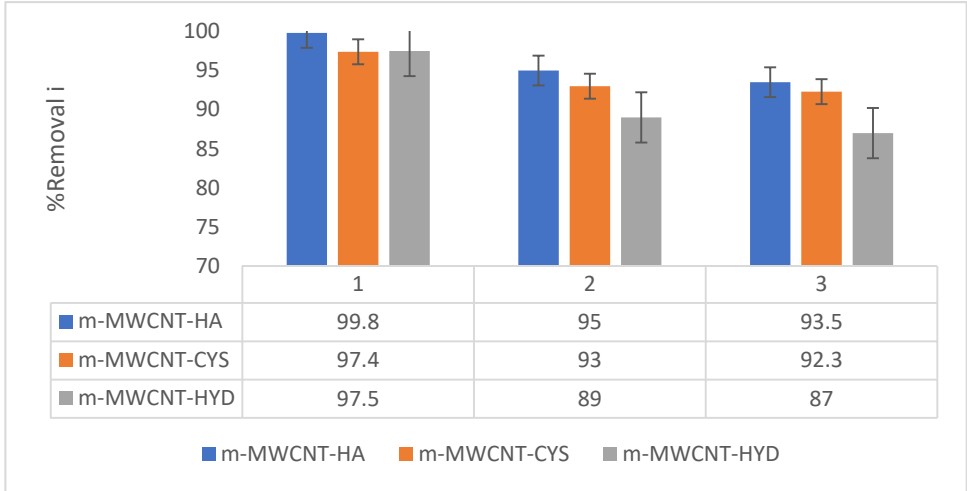

**Figure 6.** Three trials of adsorption—desorption of MWCNT using different functional groups towards removal efficiency of Pb(II).

## 3.7. Real Water Samples

The removal of metal by prepared adsorbents from real environmental water sample was studied to investigate the applicability of the proposed solid adsorbent. The results were presented in Table 5. It is clear from the table that m-MWCNT-HA, m-MWCNT-CYS, m-MWCNT-HYD has excellent efficiency towards most of the metal present in real water sample. The results obtained showed that the efficiency for the removal of lead were decreased when using real sample due to presence of dissolved organic matter and other metals. The removal efficiency decreased to about 64%, 73.1% and 67.1% when using m-MWCNT-HA, m-MWCNT-HYD and m-MWCNT-CYS respectively. This confirm the applicability of adsorbents for environmental.

**Table 5.** Effect of adsorption of metals form real water sample.

| Analyte | ppb Real Water | ppb After Treatment with m-MW CNT-HA | % Removal after Treatment with m-MW CNT-HA | ppb After Treatment with m-MW CNT-CYS | % Removal after Treatment with m-MW CNT-CYS | ppb after Treatment with m-MW CNT-HYD | % Removal after Treatment with m-MW CNT-HYD |
|---|---|---|---|---|---|---|---|
| Al | 9.2 | 1.7 | 81.5 | 2.9 | 68.4 | 0.9 | 90.2 |
| As | 1.7 | 0.7 | 58.8 | 0.66 | 61.1 | 0.23 | 86.4 |
| Ba | 29.6 | 0.1 | 99.6 | 0.5 | 98.3 | 11.3 | 61.8 |
| Be | 0.1 | 0.1 | 0 | 0.05 | 50 | 0.1 | 0 |
| B | 120.8 | 88 | 27.1 | 89.8 | 25.6 | 77.6 | 35.7 |
| Cd | 19 | 13.5 | 28.9 | 1.7 | 91 | 1.7 | 91 |
| Cr | 22 | 0.5 | 97.2 | 13.7 | 37.7 | 11 | 50 |
| Co | 15 | 5.1 | 66 | 3.1 | 79.3 | 2.2 | 85.3 |
| Cu | 2.1 | 0.99 | 52.8 | 0.8 | 61.9 | 0.11 | 94.7 |
| Fe | 520.5 | 170 | 67.5 | 77 | 85.2 | 140.5 | 73 |
| Pb | 169 | 60.7 | 64 | 45.3 | 73.1 | 55.6 | 67.1 |
| Mn | 290 | 160 | 44.8 | 35.6 | 87.7 | 40.6 | 86 |
| Ni | 40.8 | 31.9 | 21.8 | 20.1 | 50.7 | 24.5 | 39.9 |
| Se | 80.7 | 76.2 | 5.6 | 80.7 | 0 | 70.2 | 13 |
| Tl | 0.7 | 0.4 | 42.8 | 0.4 | 42.8 | 0.2 | 71.4 |
| V | 13.7 | 6.2 | 54.7 | 13.4 | 2.1 | 2.3 | 83.2 |
| Zn | 90 | 40 | 55.5 | 55.1 | 38.7 | 36 | 60 |

## 4. Conclusions

In this study, a magnetic multi-wall carbon nanotube with hydroxylamine, cystine and hydrazine was synthesis and tested for the removal of Pb(II) from water as the first example in the literature of oxidized MWCNT modified with such functionalities. The efficiency of the prepared derivatives toward Pb(II) was studied as a function of adsorbent dose, pH, metal ion initial concertation, temperature and time. The three adsorbents showed excellent efficiency toward Pb(II) and % of removal was quantitative. The highest efficiency was determined to be at room temperature and a pH of 8.0. The kinetic study revealed that the Pb(II) adsorption by the three adsorbents followed pseudo-second-order and Langmuir isotherm model. The thermodynamic analysis showed a negative free energy, indicating a spontaneous adsorption process at room temperatures. The adsorbents were regenerated by treatment with 0.1 N HCl. After third regeneration and reuse cycles, the efficiency of the regenerated adsorbents has shown a small reduction in efficiency. The presence of dissolved organic matter and other metals in real water sample has a significant effect on lead removal efficiency.

**Supplementary Materials:** The following are available online at http://www.mdpi.com/2227-9717/8/8/986/s1, Figure S1: FT-IR spectrum for MWCNTS, m-MWCNT-COOH, m-MWCNT-HA, m-MWCNT-Cys, m-MWCNT-HYD, Figure S2: Raman spectrum for MWCNTS, m-MWCNT-COOH, m-MWCNT-HA, m-MWCNT-Cys, m-MWCNT-HYD, Figure S3: SEM images of the original and the modified carbon nano tube (right) together with diameter distribution (left). (a) MWCNTs, (b) MWCNT-COOH, (c): m-MWCNT-HA, (d) m-MWCNT-CYS and (e) m-MWCNT-HYD. Scale bars: 5 μm, Figure S4: Magnetic field dependence of the magnetization measured at (a) 300K (M–H loops) of m-MWCNT-HA, m-MWCNT-CYS, m-MWCNT-HYD and MWCNT; (b) the magnification of the central area of the hysteresis loops.

**Author Contributions:** G.H.: Investigation, formal analysis, writing—original draft. S.J. and O.H.: investigation, formal analysis, writing—original draft, funding acquisition. R.B., O.D., and A.Q. assisted in sample analysis; B.K. and S.S. assisted in sample preparation. All authors have read and agreed to the published version of the manuscript.

**Funding:** This study was supported by *Middle East Desalination Research* Center (MEDRC) and Palestinian Water Authority (PWA). Part of research was funded by the German Federal Ministry of Education and Research (BMBF).

**Acknowledgments:** The authors express their thanks to the MEDRC and PWA for their financial support during this study. Other thanks go to PADUCO 2 for some financial support. Many thanks go to the department of chemistry at An-Najah National University (ANNU) for facilitating the use of their instrumentations. This work was in part supported by the research project "Palestinian German Scientific Bridge (PGSB)" carried out by the Forschungszentrum Jülich and Palestinian Academy for Science and Technology- PALAST and funded by the German Federal Ministry of Education and Research (BMBF).

**Conflicts of Interest:** The authors report no relationships that could be construed as a conflict of interest.

## Abbreviations

| | |
|---|---|
| CNTs | Carbon nanotube |
| MWCNT | Multiwall carbon nanotube |
| MWCNT-COOH | Oxidized Multiwall carbon nanotube |
| m-MWCNT-HA | Magnetized Multiwall carbon nanotube functionalized by hydroxyl amine |
| m-MWCNT-CYS | Magnetized Multiwall carbon nanotube functionalized by cysteine |
| m-MWCNT-HYD | Magnetized Multiwall carbon nanotube functionalized by hydrazine |

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
