# Peer review of "Magnetic Multiwall Carbon Nanotube Decorated with Novel Functionalities: Synthesis and Application as Adsorbents for Lead Removal from Aqueous Medium"

_processes, doi:10.3390/pr8080986_

Round 1

Reviewer 1 Report

First of all, the article is not well described. Especially the English is not precise enough. There are also errors in the description of chemistry. For example, the reaction of acid chlorides with hydroxylamine produces hydroxamic acids, not oximes. To illustrate the inaccurate English some fragments are highlighted. English deficiencies can be misleading.
Due to the shortcommings I can not recommend publishing the manuscript.

Author Response

First of all, the article is not well described. Especially the English is not precise enough. There are also errors in the description of chemistry. For example, the reaction of acid chlorides with hydroxylamine produces hydroxamic acids, not oximes. To illustrate the inaccurate English some fragments are highlighted. English deficiencies can be misleading.
Due to the shortcommings I can not recommend publishing the manuscript.

Many thanks. We do many changes. If you need further change, we can send the manuscript for MDPI language editing.

Reviewer 2 Report

The authors report the surface functionalization of oxidized MWCNTs with cysteine, hydrazine or hydroxylamine and the use of these materials for Pb(+2) removal from aqueous solution. The mechanism and the reuse of the adsorbant was also investigated. The work is of interest and results relatively clearly presented. The following key points should be considered by the authors :

  • please check reference 7. The reference section contains also many typing errors.
  • along the whole manuscript, results should be better discussed in the context of literature.
  • lines 68-70 : clarify the sentence.
  • FT-IR and Raman spectra, BET analyses, SEM images,... of functionalized MMWCNTs should be provided, for example in a supporting information.
  • there is a problem with figure 4. The graphs are overlapping the x-legend.
  • the selectivity of Pb(+2) adsorption on functionalized MWCNTs must be investigated.
  • the efficiency of the functionalized MWCNTs must be compared to other sorbents described in the literature.

Author Response

Reviewer 2

Many thanks for your comments

The authors report the surface functionalization of oxidized MWCNTs with cysteine, hydrazine or hydroxylamine and the use of these materials for Pb(+2) removal from aqueous solution. The mechanism and the reuse of the adsorbant was also investigated. The work is of interest and results relatively clearly presented. The following key points should be considered by the authors :

  • please check reference 7. The reference section contains also many typing errors. Done
  • along the whole manuscript, results should be better discussed in the context of literature.
  • lines 68-70 : clarify the sentence. Done
  • FT-IR and Raman spectra, BET analyses, SEM images,... of functionalized MMWCNTs should be provided, for example in a supporting information. We add them as supplementary doc.
  • there is a problem with figure 4. The graphs are overlapping the x-legend. Done
  • the selectivity of Pb(+2) adsorption on functionalized MWCNTs must be investigated. Done .Add as Fig 2
  • the efficiency of the functionalized MWCNTs must be compared to other sorbents described in the literature. Done

Reviewer 3 Report

The manuscript "Magnetic multiwall carbon nanotube decorated with novel functionalities: Synthesis and application as adsorbents for lead removal” by Ghadir Hanbali et al. reports an interesting approximation to the synthesis and the high adsorption performance of magnetic MWCNT. The manuscript is well structured. However, in my opinion, there are various points that could be re-considered before publication in any case:

  • In introduction section, the authors should be addressed recent papers and highlight their hypothesis, new concepts and innovations briefly.
  • The authors synthesized an efficient adsorbent for the water decontamination. However, the adsorbent recyclability after its effective lifetime, which is relatively short, is not addressed. The design of adsorbents, especially for environmentally friendly applications, should address their recyclability or integration in a circular process, minimizing the generation of residues (e.g. Advanced Science (2020) 1902447; Chemical Engineering Journal 388 (2020) 124278). It would be positive to add a brief reflection on that matter regarding the recyclability of adsorbents after its lifetime.
  • The experimental methodology section has a lack of information. More details about the synthesis must be introduced.
  • Could the authors compare these adsorbents with other processes already published in the literature?  
  • Which treatment is suitable for this study? How you optimize it? Which model or statistical tool was used? What is the applicability of this study should also be describe in results and discussion?
  • Why not include real waste contaminated with organic pollutants or other pollutants? Note that domestic, agricultural and industrial wastewater contains a combination of various pollutants such as organic pollutants, POPs, Cr(VI), and other heavy metals.
  • Replace “ml” for “mL”.
  • The captions of all the figures are not useful. For example, in Figure 1 b-e, at least, the contact time must be indicated.
  • In results and discussion, the FT-IR, FE-SEM, BET, and VSM characterizations must be included in the manuscript.
  • In Results and Discussion, the authors must provide more details of the synthesis. The morphological characterization and chemical composition of the prepared adsorbents have to be provided.
  • The morphological, FT-IR and BET characterizations after the three recycling experiments should be provided. More details about the reusability are highly required.
  • It is better to use this notation: Pb(II). The authors are not discussing about the Pb(II) speciation.
  • It is not correct the use of this notation “lead+2”.
  • More information and contextualization about the potential of these adsorbent in real conditions is required.
  • The authors highlighted the magnetic character of these adsorbents. However, some reflections about magnetic behavior have to be included in the revised version of your manuscript.
  • Conclusions should highlight the novelty of this work.

Author Response

Reviewer 3

Many thanks for your comments

The manuscript "Magnetic multiwall carbon nanotube decorated with novel functionalities: Synthesis and application as adsorbents for lead removal” by Ghadir Hanbali et al. reports an interesting approximation to the synthesis and the high adsorption performance of magnetic MWCNT. The manuscript is well structured. However, in my opinion, there are various points that could be re-considered before publication in any case:

  • In introduction section, the authors should be addressed recent papers and highlight their hypothesis, new concepts and innovations briefly. Done (Table 1 include recent studies)
  • The authors synthesized an efficient adsorbent for the water decontamination. However, the adsorbent recyclability after its effective lifetime, which is relatively short, is not addressed. The design of adsorbents, especially for environmentally friendly applications, should address their recyclability or integration in a circular process, minimizing the generation of residues (e.g. Advanced Science (2020) 1902447; Chemical Engineering Journal 388 (2020) 124278). It would be positive to add a brief reflection on that matter regarding the recyclability of adsorbents after its lifetime.

New paragraph has been added with two references, one of them you suggested.

  • The experimental methodology section has a lack of information. More details about the synthesis must be introduced. Done
  • Could the authors compare these adsorbents with other processes already published in the literature?  Yes. Summarized in table 1
  • Which treatment is suitable for this study? How you optimize it? Which model or statistical tool was used? What is the applicability of this study should also be describe in results and discussion? Adsorption experiment was carried. Each experiment was repeated 3 times the average was used when we analyzed the data.
  • Why not include real waste contaminated with organic pollutants or other pollutants? Note that domestic, agricultural and industrial wastewater contains a combination of various pollutants such as organic pollutants, POPs, Cr(VI), and other heavy metals. I apply three prepared adsorbents on real sample and the result show decrease in removal efficiency. I insert the result in manuscript. ( Sec.2.5)
  • Replace “ml” for “mL”. Done
  • The captions of all the figures are not useful. For example, in Figure 1 b-e, at least, the contact time must be indicated. In order to reduce number of Fig. I made them as sub Fig
  • In results and discussion, the FT-IR, FE-SEM, BET, and VSM characterizations must be included in the manuscript. I will send them as a supplementary doc.
  • In Results and Discussion, the authors must provide more details of the synthesis. The morphological characterization and chemical composition of the prepared adsorbents have to be provided. Done
  • The morphological, FT-IR and BET characterizations after the three recycling experiments should be provided. More details about the reusability are highly required.
  • It is better to use this notation: Pb(II). The authors are not discussing about the Pb(II) speciation. Done . changed in whole manuscript
  • It is not correct the use of this notation “lead+2”. True. Changed
  • More information and contextualization about the potential of these adsorbent in real conditions is required. I added the result
  • The authors highlighted the magnetic character of these adsorbents. However, some reflections about magnetic behavior have to be included in the revised version of your manuscript. The result of VSM added in supplementary doc.
  • Conclusions should highlight the novelty of this work. Done

Reviewer 4 Report

July 2020

Processes

Title:  Magnetic Multiwall Carbon Nanotube Decorated with novel functionalities: Synthesis and application as adsorbents for lead removal from aqueous medium

By: Ghadir Hanbali, Shehdeh Jodeh, Othman Hamed, Roland bol, Bayan Khalaf, Asma Qdemat, Subhi Samhan and Omar Dagdag

Review on Manuscript Number: Processes - 888212

Comments for the manuscript

  1. Summary

This paper presents a study on the investigation of the adsorption efficiency of multiwall carbon nanotube, after derivatization and magnetization, as a new and renewable absorbent to remove Pb2+ from waste streams.

Overall Opinion

This study is interesting and it is a contribution to the knowledge of the efficiency of nanomaterials, more specifically nanotubes in the adsorption of toxic metals. So, the research work presented and discussed in the manuscript meets the publishing objectives of Processes.

The manuscript is, in general, well structured and the presentation of data and discussion is clear. However, the section about the “Adsorbents analysis” should be illustrated with appropriate figures.

Regarding the English language, the manuscript needs revision and editing before publication, related to a few parts of the text that need to be clarified, typos in the text, and punctuation.     

A few examples are:

Line 29 – “The thermodynamic and kinetic results analysis results supported the…”

Line 37 – “Water contamination with metals is an major environmental…”

Line 81 – “…ability to adsorb Pb2 + ions in the solution..”

Lines 244 to 246 – “Raman spectroscopy shows when comparing the spectra of the oxidized MWCNT to that for MWCNT, it could be clearly noticed that the D-band of MWCNT-COOH shifted from 1360 to 1349 cm-1 and G-band from 1605 to 1600 cm-1 and this shows that a large defection was created after oxidation.” – the sentence is confusing

Line 391 – concertation

Line 397 – “…adsorbents showed no changed…”

Author Response

Reviewer 4

Many thanks for your comments

Summary

This paper presents a study on the investigation of the adsorption efficiency of multiwall carbon nanotube, after derivatization and magnetization, as a new and renewable absorbent to remove Pb2+ from waste streams.

Overall Opinion

This study is interesting and it is a contribution to the knowledge of the efficiency of nanomaterials, more specifically nanotubes in the adsorption of toxic metals. So, the research work presented and discussed in the manuscript meets the publishing objectives of Processes.

The manuscript is, in general, well structured and the presentation of data and discussion is clear. However, the section about the “Adsorbents analysis” should be illustrated with appropriate figures.

Regarding the English language, the manuscript needs revision and editing before publication, related to a few parts of the text that need to be clarified, typos in the text, and punctuation.     

A few examples are:

Line 29 – “The thermodynamic and kinetic results analysis results supported the…” Done

Line 37 – “Water contamination with metals is an major environmental…” Done

Line 81 – “…ability to adsorb Pb2 + ions in the solution..” Done

Lines 244 to 246 – “Raman spectroscopy shows when comparing the spectra of the oxidized MWCNT to that for MWCNT, it could be clearly noticed that the D-band of MWCNT-COOH shifted from 1360 to 1349 cm-1 and G-band from 1605 to 1600 cm-1 and this shows that a large defection was created after oxidation.” – the sentence is confusing, We add the Fig. as supplementary doc. Edited

Line 391 – concertation. Done

Line 397 – “…adsorbents showed no changed…” Done

Round 2

Reviewer 1 Report

Attached are two versions of the manuscript. The corrected one has some errors which are highlighted (blue). Hope, it will be easy to to do insert amendments. The original manuscript also has some suggested corrections highlighted with comments. Hope, the manuscript is now readable and could be published after spelling improvements.

The suggested changes submitted to the first round of review. are almost all included in the second version of the manuscript

Author Response

Many thanks for the acceptance

Reviewer 2 Report

The manuscript was improved by the authors.

  • the references are not in the appropriate format.
  • from my opinion, Table 1 and the related text is not well located and should be placed after the results obtained with functionalized MWCNTs.
  • the quality of the figures must be improved. The graduations on the axes are missing.
  • the authors must revise figure 2. A complexation involving the electron deficient N atoms located alpha to the C=O group is unlikely.
  • the language can still be improved.

Author Response

Rev 2 round 2

Comments and Suggestions for Authors

The manuscript was improved by the authors.

  • the references are not in the appropriate format. Done
  • from my opinion, Table 1 and the related text is not well located and should be placed after the results obtained with functionalized MWCNTs. Done
  • the quality of the figures must be improved. The graduations on the axes are missing. Done
  • the authors must revise figure 2. A complexation involving the electron deficient N atoms located alpha to the C=O group is unlikely. Modified
  • the language can still be improved. Done

Reviewer 3 Report

Accept in present form

Author Response

Many thanks for acceptance
